# Unreliable Findings Due to Miscalculations and Errors. Comment on Nally et al. The Effectiveness of School-Based Interventions on Obesity-Related Behaviours in Primary School Children: A Systematic Review and Meta-Analysis of Randomised Controlled Trials. *Children* 2021, *8*, 489

**DOI:** 10.3390/children10101660

**Published:** 2023-10-07

**Authors:** Keara Ginell, Beate Henschel, Lilian Golzarri-Arroyo, Yasaman Jamshidi-Naeini, Andrew W. Brown, David B. Allison

**Affiliations:** 1Department of Cognitive Science, Vassar College, Poughkeepsie, NY 12604, USA; 2Department of Epidemiology and Biostatistics, Indiana University School of Public Health-Bloomington, Bloomington, IN 47405, USA; 3Department of Applied Health Science, Indiana University School of Public Health-Bloomington, Bloomington, IN 47405, USA; 4Arkansas Children’s Research Institute, Little Rock, AR 72202, USA; 5Department of Biostatistics, University of Arkansas for Medical Sciences, Little Rock, AR 72205, USA

Nally et al. [1] conducted a meta-analysis to examine the effectiveness of school-based interventions in changing BMI among primary school children and reported significant differences in the change in BMI and BMI z-score between intervention and control conditions in favor of the intervention. However, the conduct of the meta-analysis as reported in Figures 3 and 4 is not consistent with the reported methods and involves invalid input data and some undisclosed analytical approaches.

First, the results in Figures 3 and 4 do not match the methods and figure captions. The captions of Figures 3 and 4 read “Forest plot for standardised mean difference of change in BMI (kg/m^2^) between intervention and control groups […]” and “Forest plot for standardised mean difference of change in BMI z-score between intervention and control groups […]” [1], respectively. Upon reviewing Adab et al. [2], Anderson et al. [3], Angelopoulos et al. [4], and Tarro et al. [5], it appears that Figure 3 does not show *standardized* mean differences of *change scores*, but rather *raw* mean differences of *post-intervention* values. The data shown in Figure 4 are post-intervention values and not change scores.

Second, of the 48 studies in Nally et al.’s [1] qualitative analysis, they report excluding 10 studies from the meta-analysis for not reporting the variance in data and other reasons. The study by Rausch Herscovici et al. [6], which has the largest contribution (35.8%) to the weighted average (Figure 3), did not report a measure of variation to calculate between-group effect sizes using standard accepted calculations. Nonetheless, it was included in the meta-analysis. Even if the study by Rausch Herscovici et al. [6] was to be included, it reported findings that were not statistically significant. Yet, Nally et al. [1] concluded statistically significant intervention effects in BMI [−1.00 (−1.15, −0.85)] and BMI z-scores [−2.93 (−3.23, −2.63)] for that study (Figures 3 and 4). Methods for including studies with incomplete information are available (e.g., imputing or estimating variance, and other approaches [7]), but no such methods were disclosed by Nally et al. [1], and Nally et al.’s [1] approach led to estimates that conflict with Rausch Herscovici et al.’s [6] conclusion about statistical significance.

Third, the data used for the study by Ford et al. [8] in Figures 3 and 4 do not match the report in the original study. Values reported in Figure 3 are implausibly small (i.e., mean of 1.085 and 1.04 for the intervention and control, respectively) to be post-intervention BMI values as reported for all other studies in Figure 3, and they do not match the change from baseline BMI values reported by Ford et al. [8]. We infer that Nally et al. [1] incorrectly used change in body mass (kg) values instead of BMI in Figure 3’s analyses and change in BMI values instead of BMI z-scores in Figure 4’s analyses. Ford et al. [8] did not report BMI z-scores; rather, they reported the z-scores for the change in BMI that would be used in a z-test to obtain a *p*-value, whereas BMI z-score is an anthropometric measure in children. Thus, the Ford et al. [8] study does not report the outcome of interest to be included in the meta-analysis in Figure 4. However, we can keep the Ford et al. [8] study in Figure 3 with corrected values for the mean difference of 0.00 (95% CI: −0.35, 0.35).

After removing Rausch Herscovici et al. [6] and replacing the values for Ford et al. [8] for Figure 3, we re-analyzed the data using a fixed-effects model (consistent with Nally et al.’s [1] methods) and show no evidence indicating significant effects for the mean difference in BMI between intervention and control groups (−0.04 kg/m2; 95% CI = −0.15, 0.06; I^2^ = 53.8%, Table 1). This contrasts with Nally et al.’s [1] conclusion that BMI and BMI z-score were significantly reduced in the intervention group. We note that given that the included studies have interventions that are expected to introduce design and exposure heterogeneity, combined with evidence of moderate to substantial statistical heterogeneity, an appropriate analysis is a random-effects model. Thus, we also re-analyzed the data using a random-effects model, which resulted in similar conclusions about intervention efficacy to our fixed-effects model (Table 1). Likewise, for Figure 4, after removing Rausch Herscovici et al. [6] and Ford et al. [8], the overall standardized mean difference in BMI z-score between intervention and control groups was no longer statistically significantly different in both the fixed- and random-effects models but with a lower heterogeneity compared to Nally et al. [1] (Table 1).

The Committee on Publication Ethics’ Retraction Guidelines state that retraction should be considered if there is “clear evidence that the findings are unreliable … as a result of major error (e.g., miscalculation or experimental error)” [9]. Although we focus herein on examples of studies that substantially affect the results, we noted other inconsistencies in reporting of the original studies’ data. We further note that there are appropriate ways to include some studies that were excluded by Nally et al. [1] because of the incomplete reporting of effects by the original investigators. The findings are therefore unreliable because of miscalculation and error, and we therefore respectfully conclude that the meta-analysis by Nally et al. [1] should be retracted.

## Figures and Tables

**Table 1 children-10-01660-t001:** Results from re-analysis after corrected data extraction and including random-effects results.

	Figure 3: Mean Differences in BMI (95% CI)	Figure 4: Standardized Mean Differences in BMI z-scores (95% CI)
Reported in Nally et al. [1] - Fixed-effects model	I^2^ = 86% −0.39 (−0.47, −0.30)	I^2^ = 96% −0.05 (−0.08, −0.02)
Re-analysis ^1^ - Fixed-effects model - Random-effects model	I^2^ = 53.8% −0.04 (−0.15, 0.06) −0.07 (−0.24, 0.11)	I^2^ = 59.3% −0.02 (−0.05, 0.01) −0.02 (−0.07, 0.03)

^1^ Re-analysis for Figure 3 excluded the Rausch Herscovici et al. [6] study and used a mean difference of 0.00 (95% CI: −0.35, 0.35) for Ford et al. [8]. Re-analysis for Figure 4 excluded both the Rausch Herscovici et al. [6] and Ford et al. [8] studies.

## Data Availability

Data from our re-analysis are readily available at https://doi.org/10.17605/OSF.IO/6Q57B (accessed on 14 July 2022).

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
