# Peer review of "Unreliable Findings Due to Miscalculations and Errors. Comment on Nally et al. The Effectiveness of School-Based Interventions on Obesity-Related Behaviours in Primary School Children: A Systematic Review and Meta-Analysis of Randomised Controlled Trials. Children 2021, 8, 489"

_children, 2023, doi:10.3390/children10101660_

Round 1

Reviewer 1 Report

The comment by Ginell et al. is clear, well written, and scientifically based on their position on the article published by Nally et al.

Author Response

We'd like to thank the reviewer for their comments.

Reviewer 2 Report

1. Nally et al. should further address the inconsistency between the pooled BMI Z-score reported in Figure 4 (-0.04, 95% CI: -0.07, -0.01) and the re-analysis result reported by Ginell et al. shown in the Table 1 (-0.02, 95% CI: -0.07, 0.03).

2. As Ginell et al. suggested, “there are appropriate ways to include some studies that were excluded by Nally et al. because of incomplete reporting of effects by the original investigators”. We have not seen the endeavor from the original investigators to include some studies by converting, for example, the measures of variation into the indicator that could be included in the meta-analyses.

3. In the Discussion, the original investigators reported that “..we found significant interaction between theory-based interventions and interventions lasting more than six months and the effectiveness of changing BMI.” It seems not clear how did the authors come to the conclusion as the results did not report P value for interaction.

4. In the abstract, please clarify the outcome indicator in the statement of “…resulted in a small positive treatment effect in the control group”.

Author Response

We thank the reviewer for their comments on the Nally et al study and agree that Nally et al should make changes.

Reviewer 3 Report

Ginell et al are to be commended for finding another important misanalysis in the literature and working to correct the record. 

Dr Alison leads a team of investigators who systematically review the literature to find articles that come to the wrong conclusions, most often through errors in statistical analyses. They identified several errors in the ms by S Nally et al. While the Editor requested a revision and resubmission from S Nally et al, I believe the correct procedure is to officially retract the original article and at the Editor's discretion accept a revised manuscript as a new submission. This identifies for future readers and systematic reviewers any mss they may have come across that retain the wrong analyses.

Alison et al are noted for the precision of their statistical analyses. It is disconcerting for S Nally et al to find a statistically significant difference in BMIz, when Ginell et al report no such difference. It is important for S Nally et al to explain in their Discussion why the difference occurred. 

Author Response

We'd like to thank the reviewer for their comments. We appreciate that our work is being recognized.

We agree to retract the original article. Additionally, we have communicated with the editor that the revised analysis is still using some incorrect data.